# Progress in Regenerative Medicine: Exploring Autologous Platelet Concentrates and Their Clinical Applications

**DOI:** 10.3390/genes14091669

**Published:** 2023-08-23

**Authors:** Laura Giannotti, Benedetta Di Chiara Stanca, Francesco Spedicato, Paola Nitti, Fabrizio Damiano, Christian Demitri, Nadia Calabriso, Maria Annunziata Carluccio, Andrea Palermo, Luisa Siculella, Eleonora Stanca

**Affiliations:** 1Department of Biological and Environmental Sciences and Technologies, University of Salento, 73100 Lecce, Italy; laura.giannotti@unisalento.it (L.G.); benedetta.dichiara@unisalento.it (B.D.C.S.); francesco.spedicato@unisalento.it (F.S.); fabrizio.damiano@unisalento.it (F.D.); eleonora.stanca@unisalento.it (E.S.); 2Department of Engineering for Innovation, University of Salento, 73100 Lecce, Italy; paola.nitti@unisalento.it (P.N.); christian.demitri@unisalento.it (C.D.); 3National Research Council (CNR), Institute of Clinical Physiology (IFC), 73100 Lecce, Italy; nadia.calabriso@ifc.cnr.it (N.C.); maria.carluccio@ifc.cnr.it (M.A.C.); 4Implant Dentistry College of Medicine and Dentistry, Birmingham B4 6BN, UK; andrea.palermo2004@libero.it

**Keywords:** autologous platelet concentrates, regenerative medicine, tissue regeneration, growth factors, stem cells

## Abstract

The goal of regenerative medicine is to achieve tissue regeneration. In the past, commonly used techniques included autologous or allogeneic transplantation and stem cell therapy, which have limitations, such as a lack of donor sites in the case of autologous transplantation and the invasiveness of stem cell harvesting. In recent years, research has, therefore, focused on new and less invasive strategies to achieve tissue regeneration. A step forward in this direction has been made with the development of autologous platelet concentrates (APCs), which are derived from the patient’s own blood. They can be classified into three generations: platelet-rich plasma (PRP), platelet-rich fibrin (PRF), and concentrated growth factors (CGFs). These APCs have different structural characteristics, depending on the distinctive preparation method, and contain platelets, leukocytes, and multiple growth factors, including those most involved in regenerative processes. The purpose of this review is to clarify the most used techniques in the field of regenerative medicine in recent years, comparing the different types of APCs and analyzing the preparation protocols, the composition of the growth factors, the level of characterization achieved, and their clinical applications to date.

## 1. Introduction

Regenerative medicine is an emerging field of medicine that was born in the 1990s.It focuses on the development of alternative therapies for the repair of cells, tissues, and organs to restore functions damaged by congenital defects, disease, trauma, or ageing. The goal is to induce self-renewal processes in damaged tissues, unlike traditional medicine, which primarily involves the replacement of damaged parts [1,2].

Historically, the gold standard of regenerative medicine has been the use of autologous transplants, where the donor and recipient are the same person, mainly due to the absence of immunogenic reactions. The clinical benefits of autologous stem cell transplantations have been observed in many clinical trials addressed to a large spectrum of disease and injury [3,4]. However, despite its high efficacy, this method has drawbacks; the autologous bone graft, for example, has limitations related to the limited quantity and availability of donor sites, pain and morbidity in the donor site, and the need for further surgery for the patient, which can lead to hematomas, infections, blood loss, and longer operative times [5]. An alternative to autologous transplantation is allogeneic transplantation, in which the donor and recipient are not the same person, but this method also has drawbacks that are mainly related to high costs and possible immune reactions leading to graft rejection [6,7].

### 1.1. Stem Cell Therapy for Tissue Regeneration

Recently, regenerative medicine has focused on research into minimally invasive techniques to regenerate damaged tissues. A breakthrough in this direction has been the use of stem cell therapy, which aims to regenerate damaged tissues and organs with new cells derived from stem cell transplantation [8,9].

In the literature, numerous studies have investigated mesenchymal stem cells (MSCs) for tissue regeneration, including bone, cartilage meniscus, tendons, ligaments, and intervertebral discs, both in vitro and in vivo [10,11]. Although stem cell therapy represents a novel approach to tissue regeneration, its clinical application is hampered by challenges, such as the limited survival and differentiation capacity of transplanted cells [12]. In addition, MSC therapy involves the collection, isolation, and in vitro expansion of these cells to ultimately enable their transplantation. This clinical use is complex due to the potential contamination of the cell culture, the overall duration of the whole process, and the high cost involved [7]. Furthermore, the risks related to tumorigenicity, pro-inflammatory properties associated with high exposure to pro-inflammatory cytokines, and the danger of developing fibrosis due to the MSC’s capability to differentiate into myofibroblasts should be taken into consideration [13]. To address these limitations, tissue engineering technology has been used to enhance the viability and proliferative potential of stem cells. Tissue engineering involves the use of a combination of cells, biomaterials, biochemical and physicochemical elements, and engineering methods to enhance or replace biological tissues [12].

### 1.2. Growth Factors and Tissue Regeneration

New strategies in regenerative medicine aim to better understand and exploit the potential of endogenous stem cells and autologous growth factors, which are specialized polypeptide molecules that bind receptors on target cells and deliver messages regarding migration, proliferation, differentiation, survival, and cell secretion [14]. The classes of growth factors most involved in tissue regeneration are considered below, including transforming growth factor beta (TGF-*β*), bone morphogenetic proteins (BMPs), vascular endothelial growth factor (VEGF), and platelet-derived growth factor (PDGF). The reported growth factors are all produced mainly by platelet activation following tissue injury (Figure 1) [15].

One of the most important growth factors in tissue repair is TGF-*β*. The TGF-*β* superfamily consists of thirty-three polypeptides, including three 25 kDa isoforms, TGF-*β*1, TGF-*β*2, and TGF-*β*3, bone morphogenetic proteins (BMPs), and activins [16].

Members of the TGF-*β* family are secreted by platelets, endothelial cells, lymphocytes, and macrophages. They act by binding as dimers to surface receptors with serine/threonine kinase activity (Figure 2). Upon binding, two type II receptors (TGFBR-II) and two type I receptors (TGFBR-I) form a stable complex in which the TGFBR-II receptors phosphorylate and activate the TGFBR-I receptors. The activated TGFBR-I receptors then phosphorylate the cytoplasmic transcription factors Smad2 and Smad3, which form a trimer with Smad4. The resulting complex translocates to the nucleus and regulates the expression of target genes [17,18,19].

TGF-*β*1, the predominant isoform, plays a crucial role in wound healing. It is involved in inflammation, angiogenesis, re-epithelialization, and connective tissue regeneration. TGF-*β*1 is also an important regulator of extracellular matrix (ECM) synthesis. It increases fibronectin and collagen gene expression, inhibits ECM degradation by metalloproteinases, and thus strengthens the matrix. TGF-*β* also plays a critical role in bone formation by increasing the chemotaxis and mitogenesis of osteoblast precursors and stimulating the deposition of the collagen matrix by osteoblasts. In addition, TGF-*β*1 upregulates VEGF, thereby promoting angiogenesis [20].

Other members of the TGF-*β* superfamily include BMPs, the largest subset of growth factors with approximately 30 different ligands. BMPs undergo multiple processing and post-transcriptional modifications before forming homodimers of two identical subunits that bind to heterodimeric type I and type II serine/threonine kinase receptors (Figure 3). The formation of this complex leads to the activation of receptor-activated Smads (RA-Smads, Smad1/Smad5/Smad8), which recruit the co-factor Smad4 and translocate to the nucleus. The activation of RA-Smads, the recruitment of Smad4, and the subsequent nuclear translocation are blocked by Smad6 and Smad7 [21].

BMPs are morphogenetic factors that do not induce cell proliferation but rather stimulate cells to differentiate and form endochondral bones in ectopic and heterotopic sites. Among the BMPs, BMP-2, -4, -6, and -7 are the most highly expressed in tissue damage [16].

The VEGF family plays a fundamental role in vasculogenesis, angiogenesis, and lymphangiogenesis and consists of VEGF-A, -B, -C, -D, and -E and placenta growth factor [22].

VEGF-A is the major member of this family and is produced by endothelial cells, keratinocytes, fibroblasts, smooth muscle cells, platelets, neutrophils, and macrophages [16,22]. It promotes vasculogenesis and angiogenesis by stimulating endothelial cell migration and proliferation. It also has a positive effect on endothelial cell survival by inducing the expression of the anti-apoptotic protein Bcl-2. In addition, VEGF factors have chemotactic effects on macrophages, and granulocytes and play a role in stimulating neurogenesis [20,22,23].

VEGF-A acts by binding two tyrosine kinase receptors, VEGFR-1 (Flt-1) and VEGFR-2 (KDR), which are located on the endothelial surface of blood vessels (Figure 4). This binding leads to the activation of several cellular signaling pathways, including phospholipase C*γ*(PL-C*γ*), phosphatidylinositol 3-kinase (PI3K), Akt, Ras, Src, and mitogen-activated protein kinases (MAPKs). The phosphorylation of PL-C*γ* stimulates Ca^2+^ release and the activation of protein kinase C (PKC), which in turn activates the Raf/MEK/ERK pathway and promotes cell proliferation. VEGF-A-activated signaling also increases cell proliferation and vascular permeability by activating endothelial nitric oxide synthase (eNOS). Finally, Src activation induces p38 MAPK activity, which increases endothelial cell migration and motility [24,25].

After an injury, activated platelets and macrophages release VEGF-A and tumor necrosis factor-alpha (TNF-*α*), which in turn induce VEGF-A expression in keratinocytes and fibroblasts. Other cytokines and growth factors act as paracrine factors, stimulating the expression of VEGF-A, including TGF-*β*1, EGF, TGF-*α*, KGF, FGF-b, PDGF-BB, and IL-1*β*. Furthermore, VEGF expression has been found to be highly upregulated in hypoxia, a hallmark of tissue damage [16,22].

The PDGF family consists of homo- and hetero-dimeric growth factors, including PDGF-AA, PDGF-AB, PDGF-BB, PDGF-CC, and PDGF-DD (Figure 5), which are abundant in platelet *α*-granules but are also present in macrophages, vascular endothelial cells, fibroblasts, and keratinocytes. These factors bind to the transmembrane tyrosine kinase receptors *α* and *β*, leading to the formation of homo- or hetero-dimers and their subsequent autophosphorylation. This triggers several signaling pathways including Src, PI3K, PLC-*γ,* and RAS [20,26].

PDGF plays a critical role in wound healing processes as it is released from platelet degranulation. It stimulates the proliferation and chemotaxis of neutrophils, macrophages, fibroblasts, and smooth muscle cells to the site of injury. It also stimulates macrophages to produce and secrete other growth factors, such as TGF-*β*. PDGF also participates in angiogenesis by inducing the expression of VEGF and VEGFR-2. Finally, PDGF has been shown to stimulate fibroblast proliferation and thus ECM production [16].

## 2. Autologous Platelet Concentrates

Further improvements in tissue regeneration have been achieved biologically using autologous growth factors derived from the patient’s own peripheral blood. The usage of blood products for non-transfusional applications is governed by stringent regulations. Within the European community, the current regulations, as set out in Directive 2002/98/EC of the European Parliament and the Council of 27 January 2003, have the primary objective of ensuring a high level of human health protection. At the European level, there is a basic principle that blood components can only be used after authorization by the national authorities responsible for transfusion activities [27].

In Italy, this principle does not allow any exceptions, and blood components for topical use are always under the jurisdiction of the transfusion service. This is true regardless of the quantity, type, and processing protocol intended for clinical use.

Conversely, in some European countries, blood components are classified as blood derivatives, whereas in others they are considered medicinal products [27].

Concentrated platelets are undoubtedly prepared by simple physical methods (such as centrifugation and separation), as long as these procedures are not used for experimental purposes or somatic cell therapy [27]. Autologous platelet concentrates (APCs) can be defined as preparations derived from the centrifugation of the patient’s whole blood. The result is a biomaterial containing a high concentration of platelets, as well as leukocytes and growth factors. In fact, platelets contain high levels of growth factors, such as PDGF, TGF-*β*1 and -*β*2, VEGF, FGF, and insulin-like growth factor-1 (IGF-1), all of which are involved in cell proliferation, matrix remodeling, and angiogenesis [28].

The efficacy of APCs in promoting wound healing and tissue regeneration has been the subject of considerable scientific interest in recent years. Due to their ability to facilitate the recruitment, proliferation, and maturation of cells involved in the regeneration of tendons, ligaments, bones, and cartilage, platelet concentrates may be beneficial for tissues with limited blood supply, slow cell turnover, and limited extracellular matrix repair [29].

They already have various applications in the medical field, including the stimulation of tissue regeneration in dentistry, plastic surgery, and implantology, the healing of refractory ulcers or burns, the repair of musculoskeletal injuries, tendons, and ligaments, and the treatment of osteoarthritis [30].

The main advantages of using platelet derivatives are their autologous nature and the simplicity of collection and preparation. They can be classified into three different generations based on their characteristics and the preparation methods required.

### 2.1. Platelet-Rich Plasma (PRP)

The first generation of platelet concentrates is called platelet-rich plasma (PRP), and its discovery dates back to 1970. PRP is derived from the patient’s whole blood and is obtained by a two-stage centrifugation process: separation and concentration (Figure 6). For the separation phase, four aliquots of 8 mL of whole blood are collected in a sterile tube containing a citrate–phosphate–dextrose anticoagulant solution centrifuged at 5600 rpm [31,32]. This results in a separation into three layers:1.The first layer contains platelet-poor plasma (PPP) and represents 40% of the total volume;2.The middle layer is called the buffy coat (BC) and contains platelets and leukocytes, making up only 5% of the total volume;3.The bottom layer is made up of red blood cells (RBCs) and represents 55% of the total volume.

At this point, the operator aspirates all the PPP and BC using a sterile syringe and transfers them to another tube without anticoagulant. This is then subjected to the next concentration step, which is achieved by centrifugation at 2400 rpm. The second phase allows the concentration of platelets at the bottom of the tube and results in three distinct layers: residual RBC at the bottom, PPP at the top, representing 80% of the total volume, and an intermediate layer of BC known as PRP [31,32]. PRP is then prepared by resuspending the platelets in an appropriate volume of plasma (2–4 mL), and then a mixture containing bovine thrombin and calcium chloride is added to activate the platelets, obtaining “activated” PRP. It appears as a gel due to the incorporation of platelets into a fibrin mesh, and only then it is ready for use [33,34].

It has been estimated that the concentration of platelets and growth factors in PRP is 3–5 times higher than in peripheral blood [35].

Based on the platelet and leukocyte content, PRP can be further divided into two categories: leukocyte-poor or pure platelet-rich plasma (P-PRP) and leukocyte- and platelet-rich plasma (L-PRP) [36]. L-PRP is obtained as described above for the PRP; in order to produce P-PRP, 11 mL of anticoagulated whole blood is centrifuged at 160 g for 10 min at room temperature to separate platelet-containing plasma from RBC and BC (rich in leukocytes). Plasma is transferred to a new tube, centrifuged, and the supernatant plasma is discarded; the precipitated platelets are resuspended in an appropriate plasma volume of about 1 mL to obtain P-PRP [37], which has a low-density fibrin network and a low or absent leukocyte content, and platelets may be damaged during preparation. On the other hand, L-PRP is characterized by a high concentration of leukocytes and platelets, while the fibrin network remains relatively loose [38].

PRP has been shown to contain numerous growth factors involved in tissue repair, but it has a significant drawback: its preparation requires the use of anticoagulants and bovine thrombin to induce fibrin polymerization, which can interfere with the physiological healing process [38]. Subsequently, in plasma rich in growth factors (PRGF), introduced by Anitua et al., animal-derived thrombin was replaced by a 10% calcium chloride solution to induce fibrin polymerization [39,40]. Also, in this case, the efficacy of leukocyte collection remains low, while platelet integrity is preserved [38].

### 2.2. Platelet-Rich Fibrin (PRF)

The second generation of platelet concentrates is platelet-rich fibrin (PRF), which was introduced by Choukroun in 2000 as an alternative to PRP. In this case, venous blood samples of 10 mL are taken without the use of anticoagulants and centrifuged at 3000 rpm for 10 min (Figure 7). At the end of this process, three layers are obtained: the lower layer consisting of RBC, the intermediate layer containing the fibrin clot, known as PRF, and the upper layer of plasma, PPP [34].

In the absence of anticoagulants, within a few minutes of contact with the tube walls, the platelets are activated, and clotting factors are released. Initially, fibrinogen is concentrated in the upper part of the tube before exogenous thrombin converts it to fibrin, which is localized in the center of the tube between the red cell layer and the acellular plasma layer [31,32].

The cellular component of the PRF includes leukocytes, platelets, macrophages, granulocytes, neutrophils, and red blood cells. These cells play a fundamental role in the process. Platelets contribute to the initial phase of healing, while leukocytes, macrophages, granulocytes, and neutrophils participate in the anti-inflammatory phase [40].

Like PRP, PRF can be further subdivided into leukocyte-poor or pure platelet-rich fibrin (P-PRF), leukocyte- and platelet-rich fibrin (L-PRF), and injectable platelet-rich fibrin (I-PRF) [36]. L-PRF has low leukocyte content but has high platelet uptake capacity and preserves platelet integrity. Like PRGF, fibrin polymerization in P-PRF is induced by calcium chloride, but the resulting fibrin matrix is denser and more stable compared to all classes of PRP. On the other hand, L-PRF is the first platelet concentrate that does not use an anticoagulant or gelling agent. This ensures the integrity of platelets and leukocytes, which remain trapped in significant quantities within the dense network of fibrin meshes that characterizes L-PRF [38]. Finally, I-PRF has been developed by reducing the centrifugation speed of PRF (700 rpm for 3 min), resulting in a liquid state PRF that can be injected into soft tissue, mucosa, or skin [41,42]. In fact, by reducing the speed and duration of centrifugation, the process of fibrin clotting can be delayed. As a result, a product consisting of fibrinogen and thrombin remains in a liquid state for approximately 20 min following centrifugation, before the eventual formation of fibrin. This characteristic renders it a suitable substance for applications in facial rejuvenation [43].

The PRF preparation protocol was subsequently modified to obtain advanced platelet-rich fibrin (A-PRF). Here, the sample is centrifuged at a lower speed for a longer time (1500 rpm for 14 min). Thanks to this modification, the fibrin clot formed in A-PRF is softer compared to P-PRF and L-PRF, and the number of platelets present in it increases [44].

### 2.3. Concentrated Growth Factors (CGFs)

The third and final generation of platelet derivatives is called concentrated growth factors (CGFs) and can also be considered a modified form of PRF that was developed by Sacco in 2006. CGFs are obtained by centrifuging the blood sample (8 mL) at alternating speeds for 13 min (2 min at 2700 rpm, 4 min at 2400 rpm, 4 min at 2700 rpm, and 3 min at 3000 rpm) using a specific centrifuge, the Medifuge MF200 (Silfradent srl, Italy) (Figure 8). At the end of this procedure, three different layers can be observed in the sample [45]: the upper layer of plasma, PPP, the middle layer containing fibrin and growth factors, CGFs, and the lower layer of RBC. The middle layer, called the CGF, can be further subdivided into the upper white part, the lower red part, and the intermediate BC [46].

The process of alternating-speed centrifugation results in the formation of a denser fibrin matrix containing more growth factors and cells compared to PRF and PRP [34,47]. The fibrin scaffold acts as a dense network containing platelets, leukocytes, and CD34+ cells [28]. Growth factors that are more abundant in CGFs include PDGF, TGF-*β*1, FGF-b, VEGF, IGF-1, EGF, and BMP. They are trapped in the fibrin matrix and released gradually, ensuring that regenerative properties are controlled and sustained over time, as well as reducing the risk of an inflammatory response associated with excessive concentrations of growth factors [45,48].

### 2.4. Comparison among the Three Generations of APCs

#### 2.4.1. Commonly Used Devices

Several devices are commonly used to prepare all types of APCs in medical and clinical settings. Here, we reported some of the devices most frequently used [49,50,51,52].

−Centrifuges:

Centrifuges are the primary tools for every APC preparation. For PRP, they use high-speed rotation to separate blood components based on their density. Some popular centrifuges for PRP include Eclipse PRP Centrifuge, the Arthrex Angel System, the Regen Lab PRP Kit, the EmCyte Pure PRP II System, and Drucker Horizon PRP Centrifuge. Instead, for PRF preparation, the devices use low-speed centrifugation for creating high-quality PRF matrices. Some examples are Choukroun PRF Centrifuge, the Process for PRF Centrifuge, and the Intra-Spin System. Regarding CGFs, the most used devices are compact centrifuges that are often utilized in smaller medical and dental clinics. The most common is Medifuge MF200, which offers customizable programs with alternating speeds, including protocols for CGFs and PRF preparation. Other examples are MediSpin, the Fibrinet System, and the Intra-Spin System.

−PRP Kits:

Many PRP preparation systems come in the form of kits that include all necessary components for blood collection, processing, and obtaining the PRP concentration. These kits often contain tubes, syringes, anticoagulants, and other accessories required for the procedure. Examples of PRP kits include those from Harvest Technologies, Arthrex, Regen Lab, and EmCyte.

−Double-Syringe Systems and Single-Syringe Systems:

These double-syringe systems simplify the PRP preparation process by automatically mixing an anticoagulant with the collected blood sample before centrifugation. They eliminate the need for manual mixing and the transfer of blood between tubes. Examples include the GPS III by Biomet Biologics and the EmCyte Pure PRP II System. Some other devices use, instead, a single-syringe mechanism for blood collection and subsequent processing. These systems may integrate centrifugation or other agitation methods to prepare PRP. Examples include the Regen Lab One PRP Kit and the Arthrex Angel System.

−PRF Boxes and Tubes:

Some PRF preparation methods involve manual manipulation of the blood sample within specialized boxes or tubes. These systems allow for the creation of fibrin matrices without the need for centrifugation. The blood is typically collected into tubes, and then the tubes are manually processed to create PRF.

−i-PRF System:

The i-PRF System is another device used to prepare PRF without anticoagulants. It employs a low-speed centrifugation process to create a fibrin matrix enriched with platelets and leukocytes. This system is commonly used in dental and oral surgery applications.

−Customizable Centrifuges:

Certain research and clinical facilities may use standard centrifuges with adjustable settings to prepare PRP or PRF according to established protocols. While these centrifuges are not designed specifically for PRP or PRF, they can be adapted for the purpose.

It is important to note that when preparing APCs, it is important to follow the specific protocols recommended by the manufacturer of the chosen device. APC preparation requires careful control of centrifugation speed, time, and other parameters to ensure the desired quality and composition of the final matrix.

#### 2.4.2. Growth Factor Release

The different preparation methods required for the different types of APCs result in differences in the concentration and kinetics of growth factor release. It is known that several factors can influence the release of growth factors from platelet concentrates, including the structure of the fibrin network and the concentration of leukocytes [27].

Masuki et al. [46] reported that CGFs and A-PRF contain similar amounts of growth factors but are higher than PRP. Qiao et al. [53] showed that the concentrations of TGF-*β*, PDGF, VEGF, and IGF-1 are similar in CGFs, PRF, and PRP, but the first two platelet concentrates have significantly higher levels of FGF-b compared to the last one. Furthermore, a study by Park et al. [54] found that a CGF contains 1.5 times more VEGF than PRP and PRF.

The release of growth factors from platelet concentrates directly affects their bioactivity.

It has been reported in the literature that PRP releases all its growth factor content within an hour of application, only participating in the initial phase of the regeneration process and not supporting the later phases. PRF, on the other hand, releases its growth factors continuously for up to 7–10 days, while the release of A-PRF appears to last up to 14 days [48]. In support of this, Schär et al. [29] also analyzed the release of growth factors from PRP and PRF over a 28-day culture period. They observed that PRP released most of its growth factors at the beginning of the culture, while PRF had a more consistent release over time. Additionally, Kobayashi et al. [55] compared the release of growth factors from PRP, PRF, and A-PRF, but over a shorter time (10 days). PRP initially releases the highest amount of growth factors, but over the entire time, PRF and A-PRF have a higher amount of growth factors released.

A study by Schär et al. [29] showed that L-PRP forms an unstable matrix that easily disintegrates during culture, leading to the immediate release of growth factors. Conversely, the more structured fibrin network formed during the progressive polymerization of L-PRF results in a sustained release of growth factors over time.

The literature reports that a CGF presents a higher quantity of growth factors compared to other platelet derivatives [46]. Several authors have shown that CGF extract contains growth factors such as VEGF, TGF-*β*1, and BMP-2, and matrix metalloproteinase (MMPs, such as MMP-2 and MMP-9) [56,57,58]. Borsani et al. [59] analyzed the release of growth factors from CGFs over an 8-day period, and their results indicated that different factors have different kinetics, indicating a programmed release of growth factors from CGFs to best support the multiple phases of the regeneration process. Honda et al. [60] reported a release of growth factors from CGFs for up to 13 days. Additionally, Stanca et al. [45] demonstrated the natural release of soluble factors from CGFs for a total period of 28 days. In fact, growth factors and MMPs are gradually released over time from CGFs, following specific release kinetics. The VEGF is slowly released up to 14 days, reaching its peak value, and then gradually decreases; TGF-*β*1 and BMP-2 are released slowly, reaching their peak release at 21 days, and the values remain high until 28 days.

Growth factors are considered essential elements in tissue regeneration and play a fundamental role in regulating processes involved in wound healing and tissue repair. Therefore, their quantities and release kinetics may be important in determining the effectiveness of CGFs. Among the variety of growth factors released by CGFs, the VEGF is a crucial molecule in tissue regeneration, as it is involved in angiogenesis and vasculogenesis processes [61]. It has been shown that the VEGF stimulates endothelial cell proliferation and promotes angiogenesis, but due to its short half-life, it has limited efficacy when administered as a protein, and higher doses often cause side effects [62]. Consequently, a CGF could constitute an ideal system that allows sustained release of the VEGF.

TGF-*β*1 has been found to be the most abundant growth factor released by CGFs over time [45]. TGF-*β*1 is a secreted protein that regulates various cellular functions, including stem cell growth, proliferation, differentiation, apoptosis, and tissue remodeling after injury [63,64]. Therefore, the release of TGF-*β*1 is beneficial at wound healing sites, particularly in the oral cavity where various cell types, including fibroblasts and osteoblasts, need to be stimulated to proliferate.

BMP-2 plays a fundamental role in bone and cartilage development by promoting the differentiation and maturation of osteoblasts [65] and it is released by platelets under acidic pH conditions, which are commonly found in wound healing sites [66,67]. According to the results obtained by Stanca et al. [45], BMP-2 is released in smaller quantities by CGFs, but the use of CGFs could enhance the regeneration process through local stimulation of BMP-2 release at the site of injury.

Furthermore, a CGF significantly releases MMP-2 and MMP-9, enzymes that degrade the matrix and are involved in inflammation and cell migration processes during wound repair [68].

In a study conducted by Isobe et al. [69], two distinct phases in the release of growth factors from CGFs were identified: an immediate phase attributed to instant release by centrifugation-activated platelets or simple diffusion and a late phase with peaks at 14 days, which could be explained by two mechanisms: the release of growth factors after degradation of the fibrin structure and the production of growth factors by the cells present in CGFs [48,70,71].

Consistently, the MMPs released in the conditioned medium by CGFs reached higher quantities compared to the initial levels extracted from CGFs, suggesting a role of CGF cells in the synthesis and secretion of such factors. MMP-2 and MMP-9 are released more rapidly than growth factors, reaching their peaks at 7 days and then gradually decreasing [45]. These results demonstrate a continuous and prolonged release of bioactive factors, suggesting that a CGF is useful in promoting the long and complex process of tissue regeneration.

#### 2.4.3. Cellular Content and Release

Concentrations of platelets in APCs were correlated with beneficial biological effects; in fact, a correlation between the concentrations of growth factors and platelets in APCs has been reported [46]. The normal concentration of platelets in the blood ranges from 150,000 to 450,000/µL. PRGF preparations contained a platelet count that was about three-fold higher than whole blood. An optimum proliferation of fibroblasts and osteoblasts [72] and increased tissue healing were demonstrated at that concentration, while platelet concentrations higher than 6-fold may inhibit tissue healing [43]. PRP preparations generally have a 4- to 8-fold higher platelet concentration than peripheral blood [43]. The concentration of platelets in A-PRF and CGFs was, respectively, 17.8-fold and 15.5-fold higher than whole blood [46]; however, the biological effect of APCs depended also on the growth factor release and the type of cells entrapped in the fibrin network.

In the literature, it has been reported that the outer surface of a CGF consists of a denser fibrin structure compared to the interior of a CGF, where a large population of activated platelets and cells is present [45]. Moreover, it has been demonstrated that the nucleated cells within CGFs exhibit a uniform distribution trapped in the fibrin network and are positive for surface markers CD34, CD45, and CD105 [45]. Indeed, the presence of different cell populations is well-known, including hematopoietic stem cells, lymphocytes, monocytes, and fibroblast-like cells [28].

Recently, Di Liddo et al. [73] reported that the platelet concentrate CPL-MB also contains autologous multipotent cells with stem cell properties.

Primary cells released from a CGF exhibit two different morphologies: one resembles fibroblast and the other is spherical. However, after a few cell passages, the predominant morphology appears to be elongated, and these cells show high levels of expression of surface markers CD105 and CD45, while CD34 is minimally detected. The primary cells of CGFs do not express the markers CD73 and CD90, which are typical of mesenchymal stem cells, but exhibit markers of multiple stem cell lineages. They express certain characteristics of mesenchymal stem cells (CD105), hematopoietic stem cells (CD45 and CD34), and endothelial progenitors (CD31 and eNOS) [45]. The literature reports that monocyte-derived cells expressing CD105, CD45, and CD14 exhibit mesenchymal cell characteristics and can differentiate into various cell lineages [74].

Additionally, CGF primary cells express genes involved in stem cell pluripotency, including Oct3/4 and Nanog [45]. The transcription factor Oct3/4 is believed to be essential for maintaining pluripotency in stem cells and is expressed in multipotent progenitor cells isolated from peripheral blood [75]. Nanog is a key factor in the self-renewal of embryonic stem cells, which maintain their pluripotency after several passages. Conversely, cells in which Nanog is not expressed show a greater propensity for differentiation [76].

Thanks to the propensity for differentiation exhibited by CGF primary cells, Stanca et al. [45] also demonstrated that they are able to differentiate into osteoblasts by expressing osteogenic characteristic markers and losing the expression of stem cell markers. In addition, Giannotti et al. [77] found that the osteogenic differentiation of CGF primary cells can be enhanced by the use of bioceramic scaffolds in silicon-doped hydroxyapatite (HA-Si).

Calabriso et al. [78] found that CGFs contained and released CD34-positive cells that expressed endothelial markers, including eNOS, VE-cadherin, VEGFR-2, and CD31, exhibiting properties of endothelial progenitor cells (EPCs). EPC-like CGF-derived cells produced multiple angiogenic molecules, including growth factors VEGF and TGF-*β*1 and matrix metalloproteinases MMP-2 and MMP-9, which are key regulators of the angiogenic process. These EPC-like cells were able to functionally integrate into the endothelial capillary-like structures of mature endothelial cells, contributing to angiogenic response [78].

#### 2.4.4. Applications in Regenerative Medicine

APCs have gained interest in various medical fields thanks to their potential to promote tissue healing, reduce inflammation, and enhance tissue regeneration. Here is an overview of the differences between their applications and the technicalities involved [79]:−Topical use (skin or mucous surfaces): topical application involves directly applying APCs to the skin or mucous membranes. This method is commonly used in dermatology, wound care, and oral surgery.−Infiltrative use (intra-tissue or intra-articular infiltration): infiltration involves injecting APCs directly into tissues or joints. This method aims to enhance tissue healing and reduce inflammation. It is often used in orthopedics, sports medicine, and pain management.−Surgical use (local application in surgical fields): APCs can be used in surgical settings to promote tissue healing and regeneration. This method is commonly employed in plastic and reconstructive surgery, oral and maxillofacial surgery, and orthopedic surgery.

It is important to note that the technicalities of each application may vary based on the specific procedure, the equipment used, and the medical professional’s preferences. The choice of concentration and activation methods can also influence the outcomes [79].

In this review we will focus on the numerous applications of APCs in the field of regenerative medicine over the past decade, mainly by surgical use, which are summarized in Table 1. One example is their utilization in facial rejuvenation. Facial aging is attributed to poor fibroblast proliferation and alterations in the ECM. Commonly used therapies include hyaluronic acid injections and adipose tissue grafts. However, hyaluronic acid is expensive and short-lived, while lipofilling may cause swelling. Growth factors could offer a new therapy by activating key factors in fibroblast proliferation and ECM remodeling [80]. It has been reported that PRP improves facial wrinkles, skin texture, and elasticity through collagen synthesis and fibroblast proliferation [81,82]. PRF appears to have a greater regenerative potential in this field compared to PRP [83]. In vitro studies have shown that CGFs may have even better effects in treating wrinkles by promoting fibroblast and endothelial cell proliferation and metabolic activity, leading to increased collagen synthesis [84].

Another important application is in the reconstruction of ears and noses. The current practice involves transplanting autologous engineered cartilage grafts, which are poorly vascularized and contain chondrocytes with low proliferative activity [85,86]. The best solution is to promote chondrocyte proliferation and ECM synthesis using platelet concentrates. PRP has been used by Fang et al. [87] on engineered cartilage grafts, and it was observed that it stimulates chondrocyte proliferation and inhibits fibrocartilage formation through the TGF-*β*/SMAD signaling pathway. CGFs have also been used to support auricular cartilage in 2019 by Chen et al. [88], who observed that they stimulate chondrocyte proliferation, migration, and ECM synthesis through the IGF-1R/PI3K/AKT pathway.

CGFs contain growth factors and a fibrin scaffold both with osteoconductive properties, making them crucial in the regeneration and repair of bone defects. Rochira et al. [89] demonstrated that CGFs can induce osteogenic differentiation in human BMSCs. Moreover, in a study by Kim et al. [90], the effects of PRP, PRF, and CGFs on the early stages of bone regeneration were compared, and no significant differences were found in their effectiveness. However, further studies have reported that both CGFs and A-PRF are better than PRP in stimulating human periosteal cell proliferation [46]. Additionally, Park et al. [54] observed that CGFs have a greater regenerative potential than PRF in the initial phase of femoral defect repair. Moreover, the combined use of CGFs and bone substitutes such as autologous bone grafts, chitosan-alginate composite hydrogels, or various types of scaffolds has shown better regenerative capabilities in the later stages of bone repair [48,91].

These results could also be attributed to the high concentration of L-glutamic acid and taurine present in CGFs, as reported by Stanca et al. [45]. It has been demonstrated that proteins and ECM biomaterials functionalized with glutamic acid-rich amino acid sequences induce osteogenic differentiation and mineralization of bone marrow stromal cells [92]. Glutamic acid residues are also known to act as nucleation points for calcium phosphate mineralization [93]. Additionally, taurine, a non-essential amino acid, has a positive effect on bone mass and influences bone metabolism [94]. Taurine is also capable of promoting the differentiation of human MSCs into osteoblasts and upregulating the expression of osteogenic markers such as osterix, RUNX2, osteopontin, and alkaline phosphatase through the ERK1/2 signaling pathway [95]. The importance of resident and circulating cells in tissue regeneration processes is also well-known [96,97].

To date, a clinically significant area of APC usage is in dental implantology due to its biocompatibility, growth factor content, and ease of being obtained [98]. The clinical use of these preparations is mainly based on the rationale that activated platelets release large quantities of growth factors and cytokines, which regulate healing processes and influence the cells involved in tissue regeneration and bone remodeling [43]. PRF has been widely used in the dental field. In vitro studies have shown that PRF can increase the expression of osteogenic differentiation markers in gingival stromal progenitor cells, periodontal ligament stem cells, alveolar bone osteoblasts, and periodontal ligament fibroblasts [99,100]. In vivo, PRF has been successfully used in maxillary sinus lifting, intraosseous defects, and dental extractions [40]. Furthermore, it has recently been observed that the application of PRF increases dental implant stability and dental bone density [101].

CGFs are also an emerging trend in periodontology and implantology. However, there are still relatively few studies evaluating the effect of CGFs on osseointegration, implant stability, survival rate, maxillary sinus augmentation, and peri-implant defects. Among these studies, Sohn et al. demonstrated that CGFs used in maxillary sinus floor augmentation induce rapid new bone formation [102]. Additionally, CGF treatment has also resulted in a better quality of the bones formed around implants, indicating implant osseointegration [103].

Palermo et al. [91] demonstrated for the first time that CGF permeation improves the success of titanium dental implants by reducing post-surgical complications. These findings agree with the results reported by Shetye et al. [104] in a recent study, which indicated that CGFs accelerate osseointegration and have a positive effect on implant stability values. However, Özveri Koyuncu et al. [105] found that the use of CGFs in dental implant surgery has no effect on implant stability compared to traditional implants. The difference between these studies lies in the fact that in the studies conducted by Özveri Koyuncu et al. [105], CGFs were always placed in the dental cavity before implant insertion. On the contrary, the technique used by Palermo et al. [91] involves the placement of an implant already permeated with CGFs inside the dental cavity.

Finally, an area of application that could prove extremely exciting and vital is the neural domain. Indeed, an in vitro study conducted by Borsani et al. [106] in 2020 reported that CGFs can induce neuronal differentiation in human neuroblastoma cells (SHSY-5Y), like commonly used differentiation inducers. This result seems to open new potential applications of CGFs in the field of neural regeneration.

**Table 1 genes-14-01669-t001:** Applications of APCs in the field of regenerative medicine.

Material	Effect of APCs	References
**Facial rejuvenation**		
PRF	Improves facial wrinkles, skin texture, and elasticity through collagen synthesis and fibroblast proliferation.	[81,82,83]
CGF	Effects treating wrinkles by promoting fibroblast and endothelial cell proliferation and metabolic activity, leading to increased collagen synthesis.	[84]
**Reconstruction of ears and noses**		
PRP on engineered cartilage grafts	This combination stimulates chondrocyte proliferation and inhibits fibrocartilage formation.	[87]
CGF	It stimulates chondrocyte proliferation, migration, and ECM synthesis.	[88]
**Regeneration and repair of bone defects**		
CGF	It can induce osteogenic differentiation in human BMSCs.	[89]
	It has a greater regenerative potential than PRF in the initial phase of femoral defect repair.	[54]
CGF and A-PRF	CGFs and A-PRF are better than PRP in stimulating human periosteal cell proliferation.	[46]
CGF + scaffold	This combination has shown better regenerative capabilities in the later stages of bone repair.	[48,91]
**Dental implantology**		
PRF	It can increase the expression of osteogenic differentiation markers in gingival stromal progenitor cells, periodontal ligament stem cells, alveolar bone osteoblasts, and periodontal ligament fibroblasts.	[99,100]
	In vivo, PRF has been successfully used in maxillary sinus lifting, intraosseous defects, and dental extractions.	[40]
	PRF increases dental implant stability and dental bone density.	[101]
CGF	CGFs, used in maxillary sinus floor augmentation, induce rapid new bone formation.	[102]
	The treatment with CGFs has resulted in a better quality of bone formed around implants, indicating implant osseointegration.	[103]
	CGF permeation improves the success of titanium dental implants by reducing post-surgical complications.	[91]
	CGFs accelerate osteointegration and have a positive effect on implant stability values.	[103]
**Neural domain**		
CGF	CGFs can induce neuronal differentiation in human neuroblastoma cells (SHSY-5Y), like commonly used differentiation inducers.	[106]

PRF, platelet-rich fibrin; CGFs, concentrated growth factors; A-PRF, advanced platelet-rich fibrin; PRP, platelet-rich plasma; APCs, autologous platelet concentrates; ECM, extracellular matrix.

## 3. Allogeneic Platelet Gel Treatments

There are certain clinical conditions that necessitate the repeated application of platelet gels over an extended period, such as large, chronic skin ulcers. However, this prolonged treatment approach may require a substantial amount of blood to be drawn, potentially compromising the patient’s hemoglobin levels. In instances like these, an alternative solution has been proposed utilizing pooled allogeneic platelet concentrates. This strategy offers the advantage of providing an essentially limitless supply of gels for repeated long-term use and introduces innovative treatment possibilities for individuals who may face challenges in undergoing autologous blood processing. For instance, it is particularly beneficial for patients who cannot undergo repeated blood draws due to various reasons, such as infants, individuals with septicemia or hematological conditions, and elderly individuals with multiple health issues. Additionally, it proves advantageous for emergency cases where prompt platelet gel treatment could be beneficial and where the time required for autologous platelet gel preparation is not feasible [107,108].

In support of this approach, in the past years, research has been conducted. Notably, a study by Greppi et al. [109] sheds light on the possibility of employing pooled allogeneic platelet concentrates to meet the demands of extended treatment regimens. Additionally, the work of Wang et al. [110] further underscores the potential of this approach.

One compelling aspect to emphasize is that the use of leucodepleted allogeneic platelet concentrates is not associated with any discernible immunological drawbacks. This facet is crucial in assuring the safety and compatibility of the treatment, offering reassurance to both patients and practitioners alike. By exploring this avenue of allogeneic platelet application, medical professionals can potentially address the challenges posed by frequent blood draws and pave the way for more sustainable, long-term treatment options [109,110].

## 4. Conclusions

This narrative review offers insights into the most recent approaches used in the field of regenerative medicine, encompassing techniques ranging from stem cell therapy to the use of APCs. The primary advantage of using APCs lies in their capacity to deliver a significant amount of growth factors to the target location, thereby stimulating angiogenesis and facilitating tissue regeneration.

This review also compares three APC generations (PRP, PRF, and CGFs), assessing their preparation protocols, growth factor composition, level of characterization achieved, and their clinical applications to date. The most effective APC seems to be the CGF, thanks to its main characteristics. It is entirely autologous, contains a higher content of growth factors compared to the other platelet derivatives, and its preparation is simple and quick. CGFs are composed of a dense fibrin scaffold, which assures a more constant delivery of growth factors (VEGF, TGF-*β*1, PDGF, FGF-b, IGF-1, EGF, and BMP), but also retains primary cells with the ability to differentiate into osteoblasts and endothelial-like cells.

Regarding the possible clinical applications of APCs in the field of regenerative medicine, the most used are CGFs, primarily in oral surgery, but also for the regeneration and repair of bone defects and facial rejuvenation.

Further studies will be necessary to gain a more comprehensive understanding of the characteristics of APCs and evaluate other potential fields of application for tissue regeneration.

## Figures and Tables

**Figure 1 genes-14-01669-f001:**
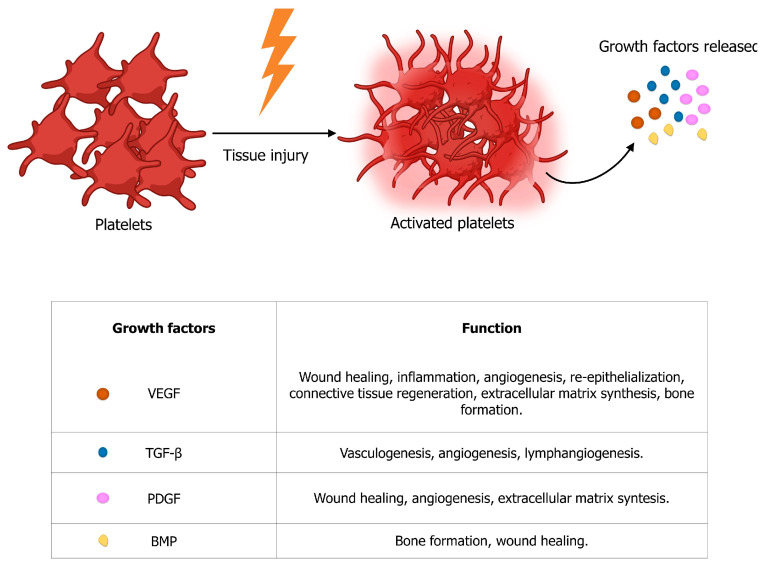
Schematic representation of platelet-released growth factors most involved in tissue regeneration: VEGF (brown circle), TGF-*β* (blue circle), PDGF (pink circle), and BMP (yellow circle).

**Figure 2 genes-14-01669-f002:**
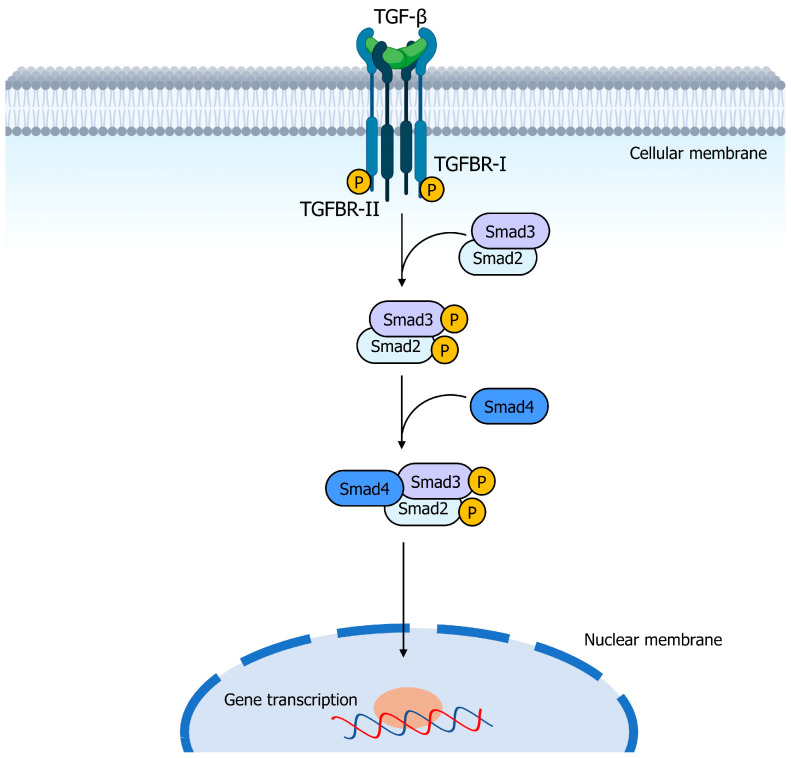
Schematic representation of the TGF-*β* signaling pathway. TGFB exerts its action by forming dimers that bind to two copies of type II receptors (TGFBR-II) and two copies of type I receptors (TGFBR-I). Upon binding, TGFBR-I receptors are phosphorylated and activated by TGFBR-II receptors. The activated TGFBR-I receptors then phosphorylate the cytoplasmic transcription factors Smad2 and Smad3, which combine with Smad4 to form a trimer. This resultant complex translocates to the nucleus and regulates target gene expression.

**Figure 3 genes-14-01669-f003:**
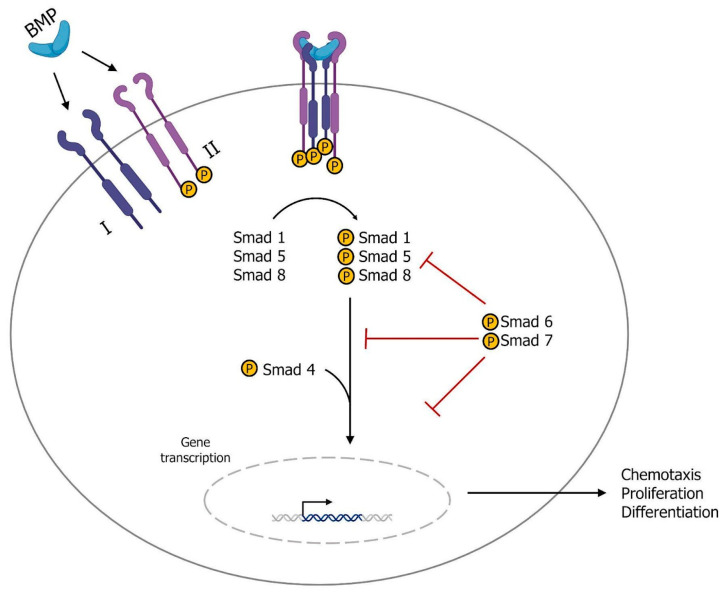
Schematic representation of the BMP signaling pathway. BMPs undergo several processing and post-transcriptional modifications prior to forming homodimers composed of two identical subunits. The homodimers then bind to heterodimeric type I and type II serine/threonine kinase receptors. The formation of this complex results in the activation of receptor-activated Smads (RA-Smads, Smad1/Smad5/Smad8), which then recruit the co-factor Smad4 and translocate into the nucleus. However, the activation of RA-Smads, the recruitment of Smad4, and the subsequent nuclear translocation can be inhibited by the presence of Smad6 and Smad7.

**Figure 4 genes-14-01669-f004:**
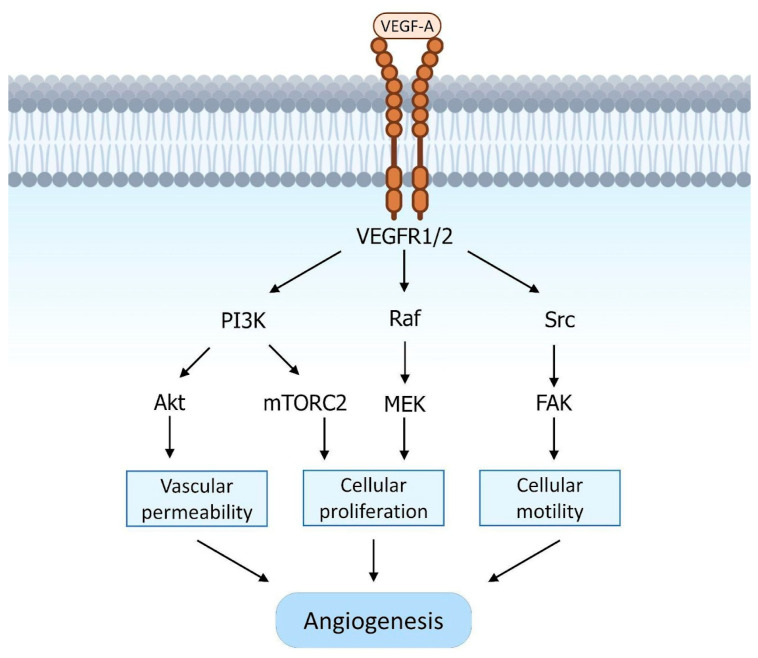
Schematic representation of the VEGF signaling pathway. VEGF-A binds to the tyrosine kinase receptors, VEGFR-1 (Flt-1) and VEGFR-2 (KDR), which are present on the endothelial surface of blood vessels. This leads to the activation of several signaling pathways, including PL-C*γ*, PI3K, Akt, Ras, Src, and MAPKs. The phosphorylation of PL-C*γ* triggers the release of Ca2+ and PKC, which further activates the Raf/MEK/ERK pathway, promoting cell proliferation. Additionally, VEGF-A-induced signaling enhances cell proliferation and increases vascular permeability by activating eNOS. Src activation also stimulates p38 MAPK activity, which enhances the migration and motility of endothelial cells.

**Figure 5 genes-14-01669-f005:**
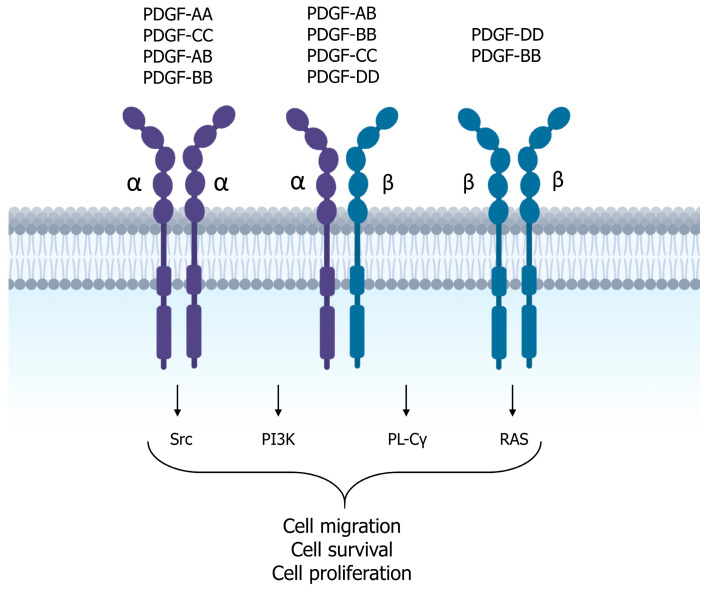
Schematic representation of the PDGF signaling pathway. The PDGF family comprises PDGF-A, PDGF-B, PDGF-C, and PDGF-D. These factors are abundant in platelet *α*-granules and can also be found in macrophages, vascular endothelial cells, fibroblasts, and keratinocytes. They interact as homo- and hetero-dimers (PDGF-AA, PDGF-AB, PDGF-BB, PDGF-CC, and PDGF-DD) with two distinct transmembrane tyrosine kinase receptors (*α* and *β*), leading to the formation of homo- or hetero-dimers and subsequent autophosphorylation.

**Figure 6 genes-14-01669-f006:**
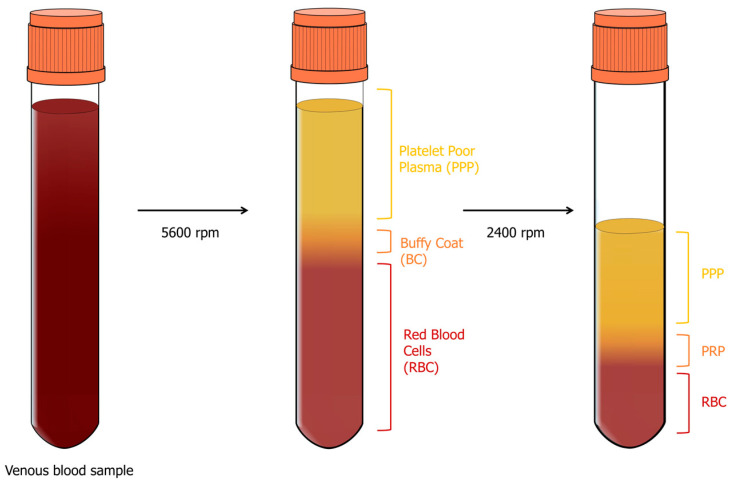
PRP preparation scheme. Venus blood is collected in a sterile tube containing a citrate–phosphate–dextrose anticoagulant solution and is centrifuged first at 5600 rpm. Three layers are formed: PPP, BC, and RBC. PPP, BC, and a small proportion of RBC are transferred to another tube without anticoagulant. This is then subjected to centrifugation at 2400 rpm, which leads to the formation of three distinct layers: residual RBC at the bottom, PPP at the top, and an intermediate layer known as PRP. To obtain “activated” PRP, the platelets are first resuspended in an appropriate volume of plasma, and then a mixture of bovine thrombin and calcium chloride is added to activate the platelets.

**Figure 7 genes-14-01669-f007:**
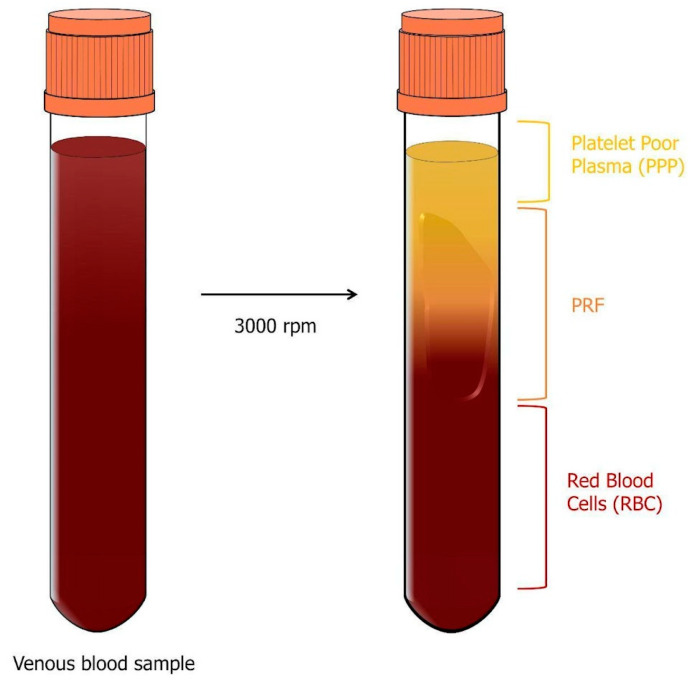
PRF preparation scheme. Venous blood samples of 10 mL are collected without the use of anticoagulants and then centrifuged at 3000 rpm for 10 min. At the conclusion of this process, three distinct layers are obtained: the bottom layer composed of RBC, the middle layer containing the fibrin clot, referred to as PRF, and the top layer consisting of plasma, PPP.

**Figure 8 genes-14-01669-f008:**
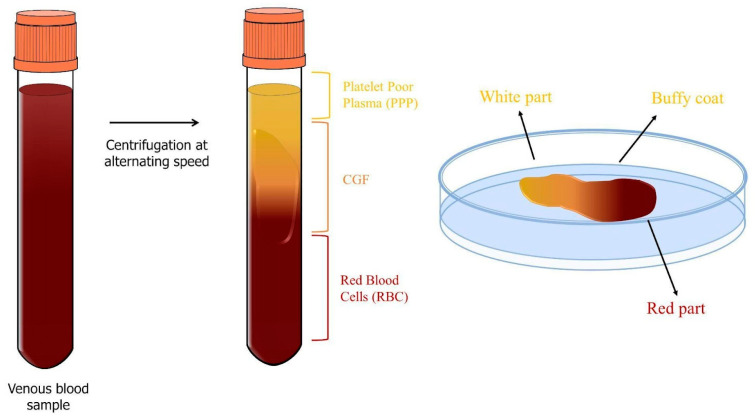
CGF preparation scheme. CGFs are obtained by subjecting the blood sample to centrifugation at alternating speeds for a total of 13 min: 2 min at 2700 rpm, 4 min at 2400 rpm, 4 min at 2700 rpm, and 3 min at 3000 rpm. After this procedure, the sample separates into three distinct layers: the upper layer consists of plasma, PPP, the middle layer contains CGFs, and the bottom layer consists of RBC. CGFs can be further subdivided into three parts: the upper white part, the lower red part, and the intermediate BC.

## Data Availability

Not applicable.

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
