# Peer review of "Progress in Regenerative Medicine: Exploring Autologous Platelet Concentrates and Their Clinical Applications"

_genes, 2023, doi:10.3390/genes14091669_

Round 1

Reviewer 1 Report

MDPI Genes-2543256

The paper by L. Giannotti and Coworkers is a review on the clinical applications of autologous platelet (PTL) preparations in regenerative medicine.

GENERAL COMMENTS

Autologous PTL concentrates and their derivatives are being used in a number of clinical settings to favor tissue repair, wound healing and cartilage or bone regeneration. Despite more than 25 years of practice, however, this topic still lacks clear clinical indications and high-quality recommendations for an effective and safe general use. A paper discussing these controversial issues and highlighting the best evidences on that matter would be therefore welcomed. 

According to its title, the present paper should focus on autologous PTL as the source of soluble factors able to provide the necessary stimuli to tissue regeneration. A long introductory chapter on mesenchimal stem cells is however present, which has no further connection with the subsequent chapters. In such a review article one may rather expect an introduction trying to describe and explain the main differences between 'conventional' autologous stem cell and PLT transplantation in tissue regeneration (i.e: the different numbers of cells to be harvested, the repeatibility of the procedure, the low therapeutic efficacy, the reduced regenerative activity and the poor viability of conventional stem cells [see: Kwon SG, et al. Biomater Res 2018 Dec 19; 22: 36]). Such a different introduction is mandatory.

When introducing autologous PLT concentrates and their clinical use, a reader would be happy to understand the differences between the application on skin or mucous surfaces (topical use), intra- tissue or intra-articular infiltration (infiltrative use) and local application in surgical fields (surgical use), along with the technicalities of the different applications. This important basic explanation should be also accompanied by a list of the evidences and recommendation grades in the various clinical settings, because the literature on that matter is often of questionable quality, with anecdotal reports and uncontrolled studies.

The non-transfusional usage of autologous PLT - at least in Europe - is a practice highly regulated by EC and governments' rules, making it available only in accredited clinical centers or in facilities under the strict supervision by a reference blood bank (see: Fiorentino S, et al. J Knee Surg 2015 Feb; 28 (1): 51-54). The usage of PLT preparations in the the various clinical settings, including aesthetic applications for wrinkles or alopecia, is also conditioned by strict local recommendations, and this issue is also to be adequately addressed.

SPECIFIC COMMENTS

Introduction, lines 41-43: This sentence is unclear. 'Limited number of donor sites' cannot be easily related to the autologous setting, and the sense of  'need for multiple surgical procedures' is also obscure. Please reword and clarify.

Chapter 1.1. Stem cell therapy. The discussion on mesenchimal stem cells is inappropriate here. Please reformat the entire chapter according to the suggestions in the general comments. Figure 1 is also very simple and can be omitted.

Chapter 1.2. Growth factors. In describing the various growth factors involved in tissue repair, please underline more clearly the ones that are mostly produced by PLT. A figure summarizing the strictly PLT-derived soluble growth factors and their respective proportions could be also of help.

Figure 5. The caption within figure 5 contains typos (Cell migrationcell, Cell survivalcell). Please amend.

Chapter 2. Autologous PLT concentrates. Please specify the different types of application and the evidence grades, as recommended in the general comments. Line 211: 'The result is a fibrin clot...' is an incorrect statement here. Please reword, by describing in order the average amount of blood drawing needed in the various settings, and the subsequent ex-vivo manipulation, also specifying the the fibrin clot is the last technical step of all preparation procedures. 

Page 11, line 336, at the end of chapter 2.3: Please describe the various possibilities to apply/inject the PLT preparations and the appropriate devices needed, with clinical examples.

Chapter 2.4.2. Cellular content. The patient's blood count requirements to allow a safe and productive draw aimed at the preparation of a functioning PLT concentrate should be indicated here, namely the minimum PLT and Hb blood levels should be indicated. Moreover, the quality assessment of the PLT concentrate before recalcification and clotting should be briefly described here.

Chapter 2.4.3. Application in regenerative medicine. Regenerative medicine is quite a vague and vast term, needing more specifications and details. The emphasis on facial wrinkles and aesthetic applications looks improper  and surprising here. A more systematic review of the major accepted clinical applications, in order of medical importance and citing  the most important 2 / 3 papers in support is needed. Organizing this point in Table form could be easy to read, and would be very appreciated by the inexperienced reader of this review.

Discussion must also take into account two other important clinical issues: 

a) There is a number of clinical conditions requiring repeated applications of PLT gels in the long term (i.e. large dull skin ulcers). This may need such a high number of large blood drawings, to jeopardize the patient's Hb level, thus hampering the use of autologous PLT in the long term. In such cases the usage of pooled allogeneic PLT concentrates, close to their expiry date, has been suggested, to provide an unlimited amount of gels for repeated use in the long term (see: Greppi N. et al, Biologicals 2011 Mar; 39(2): 73-80. and Wang S. Adv Clin Exp Med 2023 Feb 8. doi: 10.17219/acem/159088). Please add an adequate paragraph introducing this issue, also stressing the total lack of immunological drawbacks related to the leucodepleted allogeneic PLT.

b) Regulatory issues. As included in the general comments, the strict regulatory issues governing the usage of blood products for non-transfusional applications should be adequately addressed, otherwise PLT gels would seem a simple remedy to be freely used for any medical or non-medical requirement.

Language is overall good. Just a few minor amendments are required.

Author Response

The paper by L. Giannotti and Coworkers is a review on the clinical applications of autologous platelet (PTL) preparations in regenerative medicine.

GENERAL COMMENTS

  • Autologous PTL concentrates and their derivatives are being used in a number of clinical settings to favor tissue repair, wound healing and cartilage or bone regeneration. Despite more than 25 years of practice, however, this topic still lacks clear clinical indications and high-quality recommendations for an effective and safe general use. A paper discussing these controversial issues and highlighting the best evidences on that matter would be therefore welcomed. 

Response: We thank the reviewer for the valuable comments and suggestions, which can be used as a project in the future.

  • According to its title, the present paper should focus on autologous PTL as the source of soluble factors able to provide the necessary stimuli to tissue regeneration. A long introductory chapter on mesenchimal stem cells is however present, which has no further connection with the subsequent chapters. In such a review article one may rather expect an introduction trying to describe and explain the main differences between 'conventional' autologous stem cell and PLT transplantation in tissue regeneration (i.e: the different numbers of cells to be harvested, the repeatibility of the procedure, the low therapeutic efficacy, the reduced regenerative activity and the poor viability of conventional stem cells [see: Kwon SG, et al. Biomater Res 2018 Dec 19; 22: 36]). Such a different introduction is mandatory.

Response: According to the reviewer's comment, changes have been made to paragraph 1.1 (page 2 lines 66-86).

  • When introducing autologous PLT concentrates and their clinical use, a reader would be happy to understand the differences between the application on skin or mucous surfaces (topical use), intra- tissue or intra-articular infiltration (infiltrative use) and local application in surgical fields (surgical use), along with the technicalities of the different applications. This important basic explanation should be also accompanied by a list of the evidences and recommendation grades in the various clinical settings, because the literature on that matter is often of questionable quality, with anecdotal reports and uncontrolled studies.

​​Response: thanks to the reviewer for the valuable suggestion. We have added a brief introductory part to paragraph 2.4.4 regarding the different types of application of APCs in the clinical use (page 15 lines 956-985)

  • The non-transfusional usage of autologous PLT - at least in Europe - is a practice highly regulated by EC and governments' rules, making it available only in accredited clinical centers or in facilities under the strict supervision by a reference blood bank (see: Fiorentino S, et al. J Knee Surg 2015 Feb; 28 (1): 51-54). The usage of PLT preparations in the various clinical settings, including aesthetic applications for wrinkles or alopecia, is also conditioned by strict local recommendations, and this issue is also to be adequately addressed.

Response: Thanks to the reviewer's comment, we have included a note about the European directive on the use of the blood system (page 7 lines 337-419).

SPECIFIC COMMENTS

  • Introduction, lines 41-43: This sentence is unclear. 'Limited number of donor sites' cannot be easily related to the autologous setting, and the sense of  'need for multiple surgical procedures' is also obscure. Please reword and clarify.

Response: According to the reviewer’s comment, we have rephrased the sentence, clarifying its meaning (page 1 lines 42-61).

  • Chapter 1.1. Stem cell therapy. The discussion on mesenchimal stem cells is inappropriate here. Please reformat the entire chapter according to the suggestions in the general comments. Figure 1 is also very simple and can be omitted.

Response: According to the reviewer's comment, we have made changes to the chapter 1.1 Stem cell therapy, as requested by the reviewer’s general comment (page 2 lines 66-86). We have also deleted Figure 1.

  • Chapter 1.2. Growth factors. In describing the various growth factors involved in tissue repair, please underline more clearly the ones that are mostly produced by PLT. A figure summarizing the strictly PLT-derived soluble growth factors and their respective proportions could be also of help.

Response: Thanks to the reviewer for the valuable suggestion, we stated that all the reported growth factors are produced by platelets (page 2 lines 95-97). However, we added a figure (reported in the reviewed manuscript as Figure 1) summarizing the PLT-derived growth factors, as the reviewer suggested.

  • Figure 5. The caption within figure 5 contains typos (Cell migrationcell, Cell survivalcell). Please amend.

Response: We modified figure 5 by deleting typos.

  • Chapter 2. Autologous PLT concentrates. Please specify the different types of application and the evidence grades, as recommended in the general comments. Line 211: 'The result is a fibrin clot...' is an incorrect statement here. Please reword, by describing in order the average amount of blood drawing needed in the various settings, and the subsequent ex-vivo manipulation, also specifying the the fibrin clot is the last technical step of all preparation procedures. 

Response: According to the reviewer’s comment, we modified former line 211 “The result is a fibrin clot...” (now line 416) and we added the average amount of blood drawing needed for each type of APC reported (page 8 line 437; page 9 line 539; page 11 lines 611).

  • Page 11, line 336, at the end of chapter 2.3: Please describe the various possibilities to apply/inject the PLT preparations and the appropriate devices needed, with clinical examples.

Response: As already requested by the reviewer’s general comments, we added a brief paragraph describing the different types of uses of APCs (page 15 lines 956-985). We also added a paragraph comparing the different types of devices used for the preparation of PRP, PRF and CGF (page 11 lines 637-727).

  • Chapter 2.4.2. Cellular content. The patient's blood count requirements to allow a safe and productive draw aimed at the preparation of a functioning PLT concentrate should be indicated here, namely the minimum PLT and Hb blood levels should be indicated. Moreover, the quality assessment of the PLT concentrate before recalcification and clotting should be briefly described here.

​​Response: We thank the reviewer for the valuable suggestion. We modified the chapter reporting the PLT concentration required to allow a safe and productive draw aimed to the preparation of a functioning PLT concentrate (page 14 lines 855-865). To the best of our knowledge, in literature there is no data regarding the Hb blood levels of patients. Further research studies should be necessary to evaluate this important matter.

  • Chapter 2.4.3. Application in regenerative medicine. Regenerative medicine is quite a vague and vast term, needing more specifications and details. The emphasis on facial wrinkles and aesthetic applications looks improper  and surprising here. A more systematic review of the major accepted clinical applications, in order of medical importance and citing  the most important 2 / 3 papers in support is needed. Organizing this point in Table form could be easy to read, and would be very appreciated by the inexperienced reader of this review.

Response: Based on the reviewer's comment, we have attempted to provide a clearer explanation of the potential clinical applications, in the hope that these clarifications will improve the reader's understanding (lines 955-1242). We have also attempted to establish a hierarchy of importance as objectively as possible and we add a summary table (reported in the reviewed manuscript as Table 1), as suggested by the reviewer.

  • Discussion must also take into account two other important clinical issues: 

  1. a) There is a number of clinical conditions requiring repeated applications of PLT gels in the long term (i.e. large dull skin ulcers). This may need such a high number of large blood drawings, to jeopardize the patient's Hb level, thus hampering the use of autologous PLT in the long term. In such cases the usage of pooled allogeneic PLT concentrates, close to their expiry date, has been suggested, to provide an unlimited amount of gels for repeated use in the long term (see: Greppi N. et al, Biologicals 2011 Mar; 39(2): 73-80. and Wang S. Adv Clin Exp Med 2023 Feb 8. doi: 10.17219/acem/159088). Please add an adequate paragraph introducing this issue, also stressing the total lack of immunological drawbacks related to the leucodepleted allogeneic PLT.

​​Response: Thanks to the reviewer’s valuable suggestion, we have added an additional paragraph regarding the use of allogeneic platelet gels for the long term treatments or for particular clinical conditions (page 18 lines 1254-1278).

  1. b) Regulatory issues. As included in the general comments, the strict regulatory issues governing the usage of blood products for non-transfusional applications should be adequately addressed, otherwise PLT gels would seem a simple remedy to be freely used for any medical or non-medical requirement.

Response: We thank the reviewer for providing the requested insight into the current European regulations and confirm that we have addressed this point in page 7 lines 337-419 with the above guidance.

Reviewer 2 Report

In this manuscript, Giannotti et al summarize information on stem cell therapy and growth factors in tissue regeneration, compare the generation of different kind of autologous platelet concentrates and finish with a chapter about applications in regenerative medicine.

Comments:

-          - Page 3, l. 78: The authors should provide more information about the “excellent results”.

-          - Figure 2: The authors should add phosphorylation of TGFBR-I receptors into the figure.

-          - Figure 5: Color-coding next to PDGF-AA, PDGF-CC, and so on is unclear. What should it indicate?

-          - Figure 5: Typos in the bottom part, e.g. Cell migrationcell

-          - Page 9, l. 262-265: The authors have to explain how one gets P-PRP vs. L-PRP. What is the difference in the generation process?

-          - Page 10, l. 310: A-PRF is softer compared to which type of PRF?

-          - The authors mention that CGF is obtained using a specific centrifuge. But many people use also specific centrifuges to obtain I-PRF or A-PRF.

Author Response

In this manuscript, Giannotti et al summarize information on stem cell therapy and growth factors in tissue regeneration, compare the generation of different kind of autologous platelet concentrates and finish with a chapter about applications in regenerative medicine.

Comments:

1)  Page 3, l. 78: The authors should provide more information about the “excellent results”.

Response: Thanks to the reviewer’s valuable suggestion, we have changed paragraph 1.1.

2) Figure 2: The authors should add phosphorylation of TGFBR-I receptors into the figure.

Response: According to the reviewer’s comment, we modified figure 2 adding phosphorylation of TGFBR-I receptors.

3) Figure 5: Color-coding next to PDGF-AA, PDGF-CC, and so on is unclear. What should it indicate?

Response: We thank the reviewer for his comments. The different colors used in Figure 5 referred to the monomers of the different dimers of PDGF, showing the different possible combinations. To avoid ambiguity, it was decided to eliminate the colors.

4) Figure 5: Typos in the bottom part, e.g. Cell migrationcell

Response: According to the reviewer’s comment, we modified figure 5 by deleting typos.

5) Page 9, l. 262-265: The authors have to explain how one gets P-PRP vs. L-PRP. What is the difference in the generation process?

Response: We thank the reviewer for his comment. We added the difference in the generation process between P-PRP and L-PRP (page 9 lines 520-528).

6) Page 10, l. 310: A-PRF is softer compared to which type of PRF?

Response: Thanks to the reviewer’s comment, we specified (page 11 line 605) that A-PRF is softer than previously developed PRFs (L-PRF and P-PRF), except for I-PRF, which is liquid (page 10 lines 586-590).

7) The authors mention that CGF is obtained using a specific centrifuge. But many people use also specific centrifuges to obtain I-PRF or A-PRF.

Response: According to the reviewer’s comment, we added a brief paragraph describing the different types of devices needed for the preparation of PRP, PRF and CGF (page 11 lines 637-727).

Round 2

Reviewer 1 Report

The Authors have extensively reviewed their original manuscript and take into account the remarks aimed at improving it. The paper is now more coherent and informative and reads well. Just a few minor typos and style adjustments, that can be amended during the proofing phase.

Just a few minor typos and style adjustments, that can be amended during the proofing phase.

Reviewer 2 Report

-